# Quality Evaluation of Peony Petals Based on the Chromatographic Fingerprints and Simultaneous Determination of Sixteen Bioactive Constituents Using UPLC-DAD-MS/MS

**DOI:** 10.3390/molecules28237741

**Published:** 2023-11-24

**Authors:** Zhining Li, Yanni Ma, Feifei Li, Yue Wei, Lixian Zhang, Liqin Yu, Ling Chen, Xuefang Wang, Erjuan Ning, Lipan Zhang, Fayun Wang, Xiao Li, Chun Chang, Yi Fan

**Affiliations:** 1School of Chemical Engineering, Zhengzhou University, Zhengzhou 450001, China; aning072@126.com; 2Henan Academy of Sciences, Henan Napu Biotechnology Co., Ltd., Zhengzhou 450002, China; ni-2003@163.com (Y.M.); flylee5230@163.com (F.L.); lixianzhangly@126.com (L.Z.); qinshan1980@163.com (L.Y.); chenlin0000@163.com (L.C.); 15038241569@163.com (X.W.); juanjuan__2005@163.com (E.N.); 3Henan Academy of Sciences, Zhongyuan Meigu Microspectrum Technical Service (Henan) Co., Ltd., Luoyang 471002, China; weiyue2001@139.com; 4Henan Academy of Sciences, Henan Institute of Commercial Science Co., Ltd., Zhengzhou 450002, China; lipanzhang@163.com (L.Z.); 18537816421@163.com (F.W.); 5Henan Academy of Sciences, Henan High Tech Industry Co., Ltd., Zhengzhou 450002, China

**Keywords:** fingerprint analysis, UPLC-DAD-MS/MS, quality evaluation, quantitative analysis, peony flower

## Abstract

In this study, a validated quality evaluation method with peony flower fingerprint chromatogram combined with simultaneous determination of sixteen bioactive constituents was established using UPLC-DAD-MS/MS. The results demonstrated that the method was stable, reliable, and accurate. The UPLC chemical fingerprints of 12 different varieties of peonies were established and comprehensively evaluated by similarity evaluation (SE), hierarchical cluster analysis (HCA), principal component analysis (PCA), and quantification analysis. The results of SE indicated that similar chemical components were present in these samples regardless of variety, but there were significant differences in the content of chemical components and material basis characteristics. The results of HCA and PCA showed that 12 varieties of samples were divided into two groups. Four flavonoids (11, 12, 13, and 16), five monoterpenes and their glycosides (3, 4, 6, 14, and 15), three tannins (7, 9, and 10), three phenolic acids (1, 2, and 5), and one aromatic acid (8) were identified from sixteen common peaks by standards and liquid chromatography–mass spectrometry (LC–MS). The simultaneous quantification of six types of components was conducted with the 12 samples, it was found that the sum contents of analytes varied obviously for peony flower samples from different varieties. The content of flavonoids, tannins, and monoterpenes (≥19.34 mg/g) was the highest, accounting for more than 78.45% of the total compounds. The results showed that the flavonoids, tannins, and monoterpenes were considered to be the key indexes in the classification and quality assessment of peony flower. The UPLC-DAD-MS/MS method coupled with multiple compounds determination and fingerprint analysis can be effectively applied as a feature distinguishing method to evaluate the compounds in peony flower raw material for product quality assurance in the food, pharmaceutical, and cosmetic industries. Moreover, this study provides ideas for future research and the improvement of products by these industries.

## 1. Introduction

The peony (*Paeonia suffruticosa* Andr.) is a perennial deciduous shrub of the genus *Paeonia* in the Ranunculaceae family, known as the “King of Flowers”, and is distributed in most regions of China, especially in Luoyang, Heze, Tongling, Pengzhou, and other places where it is widely planted. In addition to being used as an ornamental plant, the root bark of peony is a famous traditional Chinese medicine (TCM), also known as “Dan bark”, which has the function of clearing heat and cooling blood, promoting blood circulation, and dispersing blood stasis [1]. In addition, according to The Compendium of Materia Medica, the peony flower is also a TCM material with a bitter taste and a gentle nature. It has the effect of clearing heat and detoxifying toxins and is mainly used to treat heat and dryness in the blood. Peony seeds are commonly used in folk to treat waist and leg pain. This suggests that there are also potentially functional active ingredients equivalent to the medicinal parts. Modern pharmacological studies have shown that peony has effects such as lowering blood sugar, anti-inflammatory, analgesic, bacteriostatic, regulating cardiovascular system, anti-tumor, clearing free radicals, and antioxida [2]. In China, peony seed oil was designated as a new resource food by the State Bureau of Health Supervision in 2011, and Danfeng peony flower was approved as a new food ingredient in 2013. These developments not only indicate the great potential of peonies for research and development, but also suggest that there is increasing interest among researchers in finding alternative varieties to the Danfeng peony flower.

In recent years, with the development of the peony industry, the functional and active value of peony flowers, leaves, and seeds, as byproducts of the production of Dan bark, has also attracted increasing attention and attention. Currently, the research on the chemical composition of peony primarily centers around the root, which contains phenols and their glycosides, terpenes and their glycosides, tannins, oligostilbenes, organic acids, volatile oils, trace elements, etc. [3,4]. Among them, flavonoids are a type of phenolic compound, including apigenin, quercetin, and kaempferol, that have a wide range of pharmacological activities such as antimicrobial, anti-inflammatory, antiviral, antioxidant, antitumor, and hypoglycemic [4]. The monoterpenes, particularly paeoniflorin, found in peony have various pharmacological activities such as antioxidant, antitumor, antithrombotic, antidepressant, and anticonvulsant effects [3]. Phenolic acid components have a variety of biological activities. Gallic acid and its derivatives have antibacterial, anti-inflammatory, antiviral, antioxidant, and other effects [3,5]. Tannin components have antimicrobial, antioxidant, antiviral, antitumor, and immunomodulatory effects [6]. In addition, monoterpenes and aromatic substances give peony flowers a unique flavor and have antibacterial, antioxidant, and other biological effects [7]. Peony flowers, the main byproduct of peony, are rich in various functional and active components [2] such as monoterpene glycosides, flavonoids, phenolic acids, tannins, and polysaccharides. They possess several benefits such as antioxidant, antibacterial, anti-inflammatory, and deodorizing properties, and have a large output. If this resource is fully expanded and utilized to further increase the added value of peony industrial resources, it will have great economic and social benefits. Thus, it is important to have an overall view of all the components to evaluate the quality of the medicinal plants, as many factors affect their quality and efficacy. It is necessary to establish a comprehensive and practical quality evaluation system for peony flowers to monitor their quality.

Due to the fact that in natural plants there may be hundreds of complex active components of which we have limited knowledge, it is almost impossible to identify all these substances and to carry on quantitative analysis. However, with the development of various new and efficient separation and analysis technologies, many trace components and new or difficult to separate and identify components will continue to be discovered. Chromatographic fingerprint analysis has been accepted as a strategy for quality assessment of herbal medicines and preparations by the WHO, the FDA, and China Food and Drug Administration (CFDA). It can not only comprehensively and integrally reflect the types and quantities of chemical substances in traditional Chinese medicine, but also serves as the main basis for identifying and controlling the quality of medicinal materials, providing scientific guidance for the quality evaluation of medicinal materials. It combines chromatographic techniques, such as GC [8,9] and LC [10], with sensitive and selective detectors like photodiode array detection (PDA) [11,12], evaporative light-scatter detection (ELSD) [13] and MS [14,15] to construct specific patterns of recognition for multiple compounds in herbs. The entire pattern of compounds can then be evaluated to determine not only the absence or presence of desired markers or activities but also the complete set of ratios of all detectable analytes [16]. Thus, in contrast to other methods, fingerprint analysis can reveal the total characteristics of TCMs in a relatively comprehensive way, rather than merely determining the contents of main components, which is appropriate for the features of complexity and ambiguity.

At present, there are few reports on the basic research of chemical composition and quality control of peony flowers. In particular, there are few studies comparing the chemical composition characteristics of different varieties of peony flowers. HPLC methods have been developed for the QC of peony flower, but these methods have the limitations of long analysis times and providing limited chemical information [17]. Yan et al. [18] established HPLC fingerprint of Danfeng Peony, matching 14 common peaks, and identifying 8 components. However, drying peony flowers at 50 °C could easily cause the loss and transformation of components. Zhang Guoqiang et al. [19] established a HPLC characteristic spectrum of pollen of *Paeonia rockii* from different origins and determined four components, namely, oxypaeoniflora, paeoniflorin, albiflorin, and quercetin, only including three monoterpenes and their glycosides, and one flavonoid compound. Moreover, only using the retention time of the reference substance to identify the chromatographic peak is prone to cause false-positive results, with low specificity and accuracy. LC–MS has been proven to be a powerful technique for the analysis and identification of natural compounds [20] and has drawn much attention in phytochemical analysis [21,22,23,24]. UPLC holds advantages over conventional HPLC with increased operation speed and improved sensitivity [25]. In addition, accurate mass data obtained from MS provides the elemental compositions of interested components. In addition, the fragment ions with accurate molecular compositions provide additional confirmation for the identifications of unknown compounds and provide higher selectivity and specificity for qualitative and quantification with fast and accurate analysis. The combination of UPLC and MS gives immense scope for the analysis of natural herbs.

In the present study, a rapid and reliable UPLC-DAD-MS/MS method of multiple components determination in combination with chromatographic fingerprint analysis was developed for quality evaluation of peony flowers. To our knowledge, this is the first report on the analysis of components in peony flowers using a UPLC-DAD-MS/MS technique. Four flavonoids, five monoterpenes and their glycosides, three tannins, three phenolic acids, and one aromatic acid were determined and chosen as standards for their bioactivities or high contents. A total of 22 compounds were detected and 16 components were quickly identified or tentatively characterized by comparing the UV data, accurate mass, and fragment information with the reference substance. Meanwhile, chemical fingerprints of 12 different varieties of peonies were established and comprehensively evaluated by SE, HCA, PCA, and quantification analysis. This provides a significant reference for the quality control of peony flowers and provides a research basis for further research on the functional activity value of different varieties of peony flowers.

## 2. Results and Discussion

### 2.1. Optimization of Chromatographic Conditions

The chromatographic conditions were optimized to obtain the best separation, symmetric peak shapes, as well as shorter run time by varying the column, the mobile phase, gradient program, flow rate, column temperature, and detection wavelength.

Columns such as Agilent Eclipse plus C18 column (4.6 mm × 100 mm, 3.5 μm particle size), Agilent prosell 120 EC C18 column (2.1 mm × 100 mm, 2.7 μm particle size), Agilent Eclipse SB C18 column (2.1 mm × 100 mm, 1.8 μm particle size), and Agilent prosell 120 SB C18 column (2.1 mm × 100 mm, 2.7 μm particle size) were examined and compared for their ability of separation resolution and retention, and the Agilent prosell 120 SB C18 column was confirmed to be the better choice. In particular, the peak effect of 10–12 min and 15–17 min was significant, as shown in Figure 1. As for the mobile phase, the acetonitrile–water system showed more powerful ability of separation and elution than the methanol–water system. The buffer solutions with the range of pH from 2 to 5 were evaluated, and it was found that excessive addition of formic acid (0.2%, pH = 2 to 3) inhibited the ion response value under negative mode scanning. Therefore, we used acetonitrile-0.1% formic acid aqueous solution as mobile phase for gradient elution according to the peaks shape, baseline, resolution, and ionization responses. In addition, the gradient elution method could fully eluate the extracted components, providing a strong guarantee for the establishment of the fingerprint. At the same time, it could elute more impurities, thus preventing the interference of subsequent samples, further improving the specificity and sensitivity. Because of the wide range in polarity for the analytes, the gradient elution mode was applied as shown in Section 3.2.

The column temperatures (25, 30, 35, 40 °C) were also studied in this paper, and the results showed that with the increase in column temperature, the peak time was advanced, the separation degree of adjacent peaks was decreased, and the response value of analytes was increased. Under the condition that the sensitivity was satisfied, a good chromatographic separation could be achieved at 30 °C. During the selection of flow rate, it was found through comparison that with the increase in flow rate, the analysis time would be further shortened, but the resolution of the target components would become smaller and smaller, especially for components with similar polarity (such as Cn and Rn), which were particularly prone to overlap and could not achieve an ideal separation effect. Moreover, entering a higher flow rate in the mass spectrometry would reduce the ionization efficiency. In addition, due to the high flow rate, the drying airflow and pressure also need to increase, resulting in poor reproducibility and increased costs. Therefore, the desired separation and acceptable tailing factor can be obtained at the flow rate of 0.40 mL/min without building too much backpressure on the column. 

Moreover, a full wavelength scan (190–400 nm) was performed to select the best detection wavelength (Figure 2a), and four different wavelengths (254, 230, 270, and 360 nm) were tested and compared (Figure 2b), As a result, most of the characteristic components in PPP had satisfactory sensitivity and good absorption at the wavelength of 270 nm. However, after sample determination, it was found that there were inverted peaks (negative peaks) in the chromatograms of both Haihuang and Huangguan samples (Figure 3), and the inverted peaks disappeared after removing the reference (Ref = 360, 100). This may be caused by the lower background value of the sample solution than the mobile phase absorption response value, or it may be caused by the large absorption differences due to different response levels of solvents and other solvents under different ultraviolet wavelengths. Therefore, the final choice was 270 nm, with no reference injection.

In conclusion, the optimal separation was achieved on the Agilent prosell 120 SB C18 column at 30 °C with a flow rate of 0.40 mL/min, gradient elution system of acetonitrile with 0.1% aqueous formic acid, and wavelength at 270 nm (no reference), and the separation was accomplished within 40 min.

### 2.2. Optimization of Extraction Conditions

To obtain satisfactory extraction efficiency, the extraction methods and extraction solvents were investigated. Compared with refluxing extraction, ultrasonic extraction was simpler and more economical and effective and did not cause the transformation of some unstable components due to the influence of temperature, and then used in further experiments. 

For extraction solvents, aqueous methanol and ethanol solutions were tested. Firstly, in this study, four methanol aqueous solutions with different volume concentrations (30%, 50%, 70%, and 90%) were selected for extraction, and the results showed that (Figure 4) the extraction values of most targets increased gradually with increased methanol concentration. Too high methanol concentration (≥90%) did not benefit efficient extraction (ingredients such as 5G). Then, 70% ethanol aqueous solution with lower toxicity was selected as the extraction solvent to explore. It was discovered that the signal response values of Ee and 5G components were higher than 70% methanol aqueous solution, but the Me component was not extracted. In order to extract more components from peony petal and more comprehensively reflect the chemical composition and material basis of peony flower, the extraction solvents with different volume ratios of methanol/ethanol/water (6:1:3, 5:2:3, 4:3:3, 3:4:3, 2:5:3, and 1:6:3, *v*/*v*/*v*) were investigated to obtain optimal extraction conditions by comparing the numbers, areas and resolution of the chromatographic peaks. The results showed that (Figure 5) with the increase in the ethanol solvent ratio, the response of components such as Rn, 5G, and Asn showed first an increase and then a decrease, while the response of components such as Cn showed first a decrease and then an increase; Me gradually decreased, and Ee gradually increased. According to the UPLC consequences, economically and environmentally friendly, a ratio of 4:3:3 was chosen, the extraction component response signal was more appropriate, thus the chromatographic fingerprint established could reflect the quality of peony flower more comprehensively, and the selection of methanol with a higher proportion could reduce the background interference in the mass spectrum. The investigation results of extraction time (15, 30, and 45 min) suggested that longer period (>30 min) of ultrasonication did not increase the contents significantly. The optimal extraction conditions for PPP were established as follows: 0.5 g of sample was ultrasonic extracted with the three mixed solutions of methanol/ethanol/water (4:3:3, *v*/*v*/*v*) at room temperature for 30 min was effective and convenient for the analytes’ extraction.

### 2.3. Fingerprints

#### 2.3.1. Method Validation of Fingerprints

According to the guideline of CFDA [26] and some reports [11,12,16], the method was validated for injection precision, repeatability, and sample stability. The characteristic peak 11 was selected as a reference to calculate the relative retention time (RRT) and relative peak area (RPA) of each characteristic peak. The RSDs of the RRT and RPA of characteristic peaks were used to reflect the injection precision, repeatability, and sample stability, respectively.

The injection precision was evaluated by successive analysis of the same sample solution six times. The results demonstrated that the RSDs of injection precision were in the range of 0.03–0.25% for the RRT and 0.14–1.55% for the RPA. The repeatability was evaluated with six independently prepared sample solutions. The RSDs of repeatability were in the range of 0.06–0.20% for the RRT and 1.04–4.04% for the RPA. The sample stability was evaluated by analyzing the same solution at different times (0, 1, 2, 4, 8, 12, and 24 h). The RSDs of sample stability were below 0.20% for the RRT and 4.67% for the RPA. These results confirmed that the method of UPLC for the fingerprint analysis was valid and satisfactory [16,26].

#### 2.3.2. Fingerprint Analysis

The development and validated UPLC method with DAD detector was applied to investigate the fingerprint chromatograms of 12 PPP samples with different varieties. Load the DAD chromatogram of the test solution into the similarity evaluation software for the chromatographic fingerprint of traditional Chinese medicine (version 2012) [27]. Data cutting was performed on the atlas with a start time of 2 and an end time of 33. Using S1 as the reference characteristic fingerprint, the UPLC-DAD reference characteristic fingerprint of S1 sample (Figure 6a) and the UPLC-DAD characteristic fingerprint of 12 PPP samples (Figure 6b) were generated through multi-point correction and full spectrum peak matching. The fingerprint chromatograms revealed the presence of approximately 22 common peaks in the chromatogram of PPP. According to the DAD chromatogram, UV spectra and MS mass spectrum of the reference solution, 16 common peaks were further identified as follows: 1, Gd.; 2, Me.; 3, Oa.; 4, Pn.; 5, Ee.; 6, Aln.; 7, 3G.; 8, Bd.; 9, 4G.; 10, 5G.; 11, Asn.; 12, Cn.; 13, Rn.; 14, Mc.; 15, Bn.; and 16, Kl.

The consequences of similarity evaluation were shown in Table 1. It was revealed that the samples with high similarity to S1 samples were S3 (0.978), S4 (0.953), S9 (0.946), and S2 (0.905) samples, which reached above 0.9, indicating a strong regularity between the various spectral peaks. The remaining samples had a lower similarity to S1, with the lowest similarity to S1 being S11 (0.459), S12 (0.653), and S10 (0.744) samples. The samples with high similarity to S9 (Danfeng) were S2 (Flesh Lotus, 0.985) and S3 (Xiangyu, 0.979), indicating that similar chemical components were present in these samples regardless of variety. Thus, it could be identified that there were significant differences in the chemical composition and material basis of peony petals among different varieties. In the later stage, the content of the characteristic components of the sample would be accurately measured to further compare the differences between them.

#### 2.3.3. Hierarchical Clustering Analysis (HCA)

The cluster analysis of 12 varieties PPP samples was performed by SPSS 26.0 soft-ware based on the similarities from 12 varieties of PPP samples. The input table of the HCA is listed in Table 1. The dendrogram of HCA was shown in Figure 7. It was clear that the 12 samples were classified into two groups (G1, G2,) at the squared Euclidean distance of 12.5. G1 (including S1–S9) mainly coming from the varieties of Zhao Fen, Flesh Hibiscus, Xiangyu, Jingyu, Xugang, King of Flowers, Haihuang, Huangguan, and Danfeng. G2 (including S10, S11, and S12) mainly coming from the varieties of Black Lady, Luoyang Red, and First Case Red. In the squared Euclidean distance of 5.0, G1 was further divided into subgroup 1a (S1, S2, S3, S4, and S9), subgroup 1b (S7, S8, S5 and S6), G2 was further divided into and subgroup 2a (S10 and S12) and subgroup 2b (S11). The result of HCA was consistent with that of the similarity evaluation. This denoted that HCA was an efficient way to identify PPP from different varieties.

#### 2.3.4. Principal Component Analysis (PCA)

PCA was also performed on the RPA to ensure that all elements equally influenced the results [28,29]. Through the eigenvalues and percentage variance determined for each PC (Table 2), we found that the first eight PCs explained 94.3% of the relevant information. Two principal components (PC1 and PC2) accounting for 28.970%, 17.387% of the total variance, respectively, were selected to represent the total variable information based on eigen values > 1. The first two principal components occupied 46.36% of the total variable, which represented almost half of the raw data. This indicated that there were varying degrees of correlation and differences between different species and genera of the same family of plants. Figure 8 that reported the PCA plot showed the discrimination of the different samples. The 12 samples were divided into two groups (G1 and G2). S1–S9 was classified as G1, and G2 included S10, S11, and S12. The classification results were consistent with those of the similarity evaluation and HCA. Among them, there were significant differences within the first group (G1). At the same time, combined with the analysis of the flower color perspective, S7 and S8 were similar and the flower color was yellow, S3 and S9 had a small difference and the flower color was white, with pink between them. S1 and S2 had the smallest difference and the flower color was pink. The second group (G2) had a small difference, but S11 (deep fuchsia) showed a discrete trend compared to S10 and S12 (fuchsia), indicating that Luoyang Red Peony still had significant differences compared to Black Lady and First Case Red Peony, and the results were consistent with their flower colors. 

### 2.4. Quantitative Determination in UPLC-MS/MS

The concentrations of the sixteen components, including Gd, Me, Oa, Pn, Ee, Aln, 3G, Bd, 4G, 5G, Asd, Cn, Rn, Mc, Bn, and Kl were quantitatively determined through UPLC-MS/MS. 

#### 2.4.1. Specificity Inspection

Under the selected chromatographic and mass spectrometric conditions, the typical ion flow chromatograms of the sixteen components in mixed standard samples and samples are shown in Figure 9 and Figure 10. It could be seen from the figure that the retention times of Gd, Me, Oa, Pn, Ee, Aln, 3G, Bd, 4G, 5G, Asd, Cn, Rn, Mc, Bn, and Kl were 2.236 min, 6.653 min, 8.050 min, 10.188 min, 10.484 min, 11.310 min, 12.356 min, 13.627 min, 15.647 min, 15.965 min, 16.030 min, 16.383 min, 16.470 min, 19.160 min, 20.371 min, and 23.612 min, respectively, and there was no interference at the target peak and no interference from other components in the sample. The theoretical plate numbers (N) of the sixteen components are 8510, 223,200, 303,370, 622,710, 100,150, 567,830, 251,840, 469,070, 985,470, 1,266,220, 1,268,470, 1,288,490, 1,502,250, 1,630,930, 1,416,510, and 3,829,190, respectively. Except for Gd from peak 1, the N values of other components were far greater than 7000. At the same time, the injection amount of the method was small, reducing the chromatographic peak width, and further increasing column efficiency. This shows that the method and ultra-high performance chromatographic column used in this study had high column efficiency, strong separation ability, and good separation effect. In addition, the MS detector was further qualitative and more specific using the MRM mode for scanning, which made qualitative and quantitative analysis more accurate and had a stronger system adaptability. Thus, this method had good specificity. 

#### 2.4.2. Method Validation of Quantitative Determination

The developed method was validated for its specificity, linearity, LODs and LOQs, injection precision, repeatability, sample stability, and accuracy. All calibration curves were plotted based on linear regression analysis of the integrated peak areas (*Y*) versus concentrations (*X*, μg/mL) of identified constituents in the standard solutions. The regression equations, correlation coefficients (*R*), and concentration ranges are shown in Table 3. From the regression equation, sixteen reference substances showed a good linear relationship within a certain concentration range. 

The injection precision test was performed by successive analysis of the same sample solution six times. The RSD values of the peak area were within 0.12–1.10% (Table 4). To evaluate the repeatability, six independent samples from the same batch (S1) were prepared and analyzed. The RSD values of the content were in range of 0.39–3.14% (Table 5), indicating the excellent reproducibility of the method.

The recovery of the assay was evaluated by spiking the extracts with an exact amount of each reference compound, which was prepared in sextuplicate. Detailed procedure of spiked samples and average recoveries for analytes are displayed in Table 6. According to the data, the recoveries of this method were 100.79–101.17%, 98.99–102.30%, 95.68–101.30%, 99.70–107.73%, 99.69–103.32%, 99.54–100.95%, 105.69–109.19%, 93.06–98.32%, 96.43–96.77%, 97.11–99.56%, 98.99–99.97%, 95.60–99.21%, 99.68–103.76%, 100.84–111.22%, 100.35–105.34%, and 103.58–115.22%, respectively. The RSDs ranged from 0.89% to 7.67%, all of which were less than 10%, and the method met the requirements of analysis and determination. 

For sample stability, the same sample solution was analyzed after preparation for 0, 1, 2, 4, 8, 12 and 24 h at room temperature. The RSD values of the peak area were less than 7%. It indicated that the stability of the test solution was good within 24 h, as shown in Table 7. The results showed that this method could ensure the reliability and accuracy of the determination results. 

#### 2.4.3. Quantitative Determination of Identified Ingredients

The developed quantitative method was applied to simultaneous determination of the sixteen active ingredients present in PPP samples of 12 varieties. The contents were calculated by the calibration curves from three parallel determinations of each sample. It was found that the contents of each analyte were various among the different samples with ranges as follows (Table 8): 0.98–3.80 mg/g (Gd), 0.30–3.34 mg/g (Me), 0.19–1.75 mg/g (Oa), 0.012–0.32 mg/g (Pn), 0.20–0.62 mg/g (Ee), 0.42–5.61 mg/g (Aln), 0.014–0.044 mg/g (3G), 0.32–2.02 mg/g (Bd), 0.36–1.23 mg/g (4G), 5.23–14.12 mg/g (5G), 0.75–7.69 mg/g (Asn), 1.32–4.77 mg/g (Cn), 4.31–9.48 mg/g (Rn), 0.0049–0.20 mg/g (Mc), 0.041–0.86 mg/g (Bn), and 0.012–0.74 mg/g (Kl). The bar graph of the average contents of six types of components from sixteen analytes in PPP samples of 12 varieties is shown in Figure 11. Combined with the HCA and PCA results (Figure 7 and Figure 8), it was found that the sum contents of analytes varied obviously for PPP samples from different varieties. The content of flavonoids, tannins, and monoterpenes (≥19.34 mg/g) was the highest, accounting for more than 78.45% of the total compounds. The contents of phenolic acids compounds varied slightly among 12 varieties, which was helpless for the quality monitoring of PPP. The ratio between the content of flavonoids and monoterpenes in PPP samples with fuchsia from group II was close to one, the group I was greater than 2. The reasons for the differences may be related to the synthesis and transfer of secondary metabolites during the growth process and among different species of peonies. Furthermore, whether this result is related to pollen collection and storage, as well as whether the stamens and flower centers also contain such components, remains to be further studied and compared. Therefore, simultaneously detecting the proportions of these compounds aforementioned is necessary, the quality of ppp samples can be completely evaluated and controlled not only qualitatively but also quantitatively at only one time injection.

## 3. Materials and Methods

### 3.1. Chemicals and Reagents

Standards of gallic acid (Gd), ethyl gallate (Ee), methyl gallate (Me), paeoniflorin (Pn), albiflorin (Aln), apigenin-7-O-β-D-glucopyranoside (cosmosiin, Cn), oxypaeoniflora (Oa), mudanpiosideC (Mc), benzoyloxypaeoniflorin (Bn), and rhoifolin (Rn) (purity ≥ 98%, Figure 12) were purchased from Chengdu Pufeide Biotechnology Co., Ltd. (Sichuan, China). Standards of 1,2,3,4,6-O-pentagalloyl glucose (5G, purity 99.33%), 1,3,6-tri-O-galloyl-beta-D-glucose (3G, purity 98.41%), and astragalin (Asn, purity 99.02%) (Figure 12) were purchased from Chengdu Dester Biotechnology Co., Ltd. (Sichuan, China). 1,2,3,6-O-tetragalloyl glucose (4G, purity 99.9%) (Figure 12) was purchased from Chengdu Phyto-Standard Pure Biotechnology Co., Ltd. (Sichuan, China). Benzoic acid (Bd, purity 99.9%) and kaempferol (Kl, purity 95.5%) (Figure 12) were purchased from China Food and Drug Control Institute (Beijing, China). Acetonitrile, methanol, and formic acid of LC–MS grade were purchased from Merck (Darmstadt, Germany). Ultrapure water with resistivity above 18 Mα cm was obtained by a Milli-Q water purification system (Millipore, Billerica, MA, USA), and the other reagents were of analytical grade and purchased from Beijing Chemical Works (Beijing, China). Twelve kinds of raw material samples of peony flower were collected from Luoyang, Henan provinces of China, and their names are: Zhao Fen, Flesh Hibiscus, Xiangyu, Jingyu, Xugang, King of Flowers, Haihuang, Huangguan, Danfeng, Black Lady, Luoyang Red, and First Case Red (the numbers were sequentially marked as S1–S12). All the crude drugs were authenticated by Professor Suiqing Chen of Pharmacognosy of Henan University of Chinese Medicine.

### 3.2. Chromatographic Analysis

UPLC analysis was performed on an Agilent1290 Infinity II system (Agilent, CA, USA) equipped with a diode array detector (DAD), which was connected to Agilent Chemstation B.08 software. All separations were carried out on an Agilent prosell 120 SB C18 column (2.1 mm × 100 mm, 2.7 μm particle size) at a flow rate of 0.40 mL/min, maintained at 30 °C. A binary gradient elution system composed of acetonitrile (solvent A) and 0.1% aqueous formic acid (*v*/*v*, solvent B) was applied as follows: 0–8 min, 2–12% A; 8–11 min, 12–16% A; 11–15 min, 16–20% A; 15–20 min, 20–25% A; 20–24 min, 25–45% A; 24–30 min, 45–65% A; 30–33 min, 65–90% A; 33–36 min, 90–2% A; and 36–40 min, 2% A. DAD was set to scan from 190 to 400 nm, and 270 nm (no reference) was used as detection wavelength with the sample injection volume of 2 μL.

### 3.3. UPLC–QQQ-MS Analysis

UPLC–ESI-QQQ-MS analysis was carried out on an Agilent1290 Infinity II system (Agilent, USA) in combination with a 6460 Triple Quad MS mass spectrometer which was equipped with an ESI source, using Agilent Mass Hunter workstation control B.08 software for data acquisition. The chromatographic conditions were as described above. The MS/MS spectra were obtained by CID with high-purity nitrogen (N_2_) as collision gas after choosing the precursor ions and product ions of interest. ESI Mass spectra were acquired over the range from *m*/*z* 100 to 1000 in positive mode. Negative mode was also done when necessary to obtain further confirmation. Nitrogen was used as the drying gas at a flow rate of 11.0 L/min and the drying gas temperature was 350 °C; the nebulizing gas (N_2_) was set at 40 psi and the capillary voltage was set at 4500 V (ESI^+^) and 3500 V (ESI^−^), respectively. The data were analyzed using Mass Hunter Data Analysis B.07 software. The ion pairs, residence time, collision energy, and fragmentation voltage used for qualitative and quantitative analysis of 16 components detected by MRM method were listed in Table 9. 

### 3.4. Preparation of Standard Solutions

Reference compounds of Gd, Me, Oa, Pn, Ee, Aln, 3G, Bd, 4G, 5G, Asn, Cn, Rn, Mc, Bn, and Kl were accurately weighed and dissolved in chromatographic methanol to yield the mixed standard solution with concentrations of 8.52, 9.12, 7.72, 8.32, 8.44, 8.04, 7.96, 8.20, 8.80, 8.08, 8.36, 8.16, 8.28, 8.72, 7.68, and 7.80 μg/mL, respectively, and stored away from light at 4 °C.

### 3.5. Preparation of Peony Petal Powder (PPP)

Took 12 varieties of peony flowers, separated the stamens and calyxes, put them on A4 paper, respectively, and freeze dried them in a freeze dryer for 72 h (−55–80 °C, vacuum < 10 Pa). After taking them out, peony petal samples were comminuted with a mill to pass through 24-mesh sieve to obtain peony pollen samples and then put it in a sealed bag for standby.

### 3.6. Analytical Sample Preparation

The accurately weighed powder (about 0.5 g) was extracted by ultrasonication for 30 min with 50 mL methanol-ethanol mixed aqueous solution (4:3:3, *v*/*v*/*v*) in a 100 mL volumetric flask. During the ultrasonic process, ice was put in to keep it at room temperature (25 °C) for ultrasonic extraction as much as possible. After cooling, the extracted solution was added to the original weight. A volume of 2 mL of the solution was filtered through a 0.22 μm PTFE filter before analysis.

### 3.7. Validation of UPLC–MS/MS Method

The newly developed UPLC–MS/MS method was validated in terms of linearity, precision, stability, repeatability, and accuracy. For the calibration curves, each concentration of the mixed standard solution was injected in triplicate. Calibration curves were established by plotting the peak area versus concentration of each analyte. The LOD and LOQ for each standard were estimated at S/N of 3 and 10, respectively. The precision of the method was evaluated by continuous analysis of six replicates of the same sample solution within one day. To confirm the repeatability, six replicates of the same samples were extracted and analyzed in a single day. Sample stability was monitored by analyzing the same sample solution at different time points (0, 1, 2, 4, 8, 12, 24, and 48 h). Recovery was investigated to check the accuracy of the method. Three different concentrations of mixed standard solutions were spiked into the same sample. Then, the samples were extracted and analyzed using the proposed method. Three replicates were performed for the test. Variations were expressed by RSD in all four tests above.

### 3.8. Establishment of Fingerprint Chromatogram and Simultaneous Determination

Twelve varieties of peonies (marked as S1, S2, S3, S4, S5, S6, S7, S8, S9, S10, S11, and S12) were analyzed under the chromatographic condition of Section 3.2 for establishing UPLC fingerprint chromatogram. The relative retention time and relative peak area were accurately calculated by similarity evaluation system software for chromatographic fingerprint of TCM as weight parameters to evaluate the fingerprint chromatogram. The contents of sixteen analytes in 12 varieties of peonies samples were simultaneously determined from the corresponding calibration curves by UPLC–MS/MS.

### 3.9. Data Analysis

Similarity analysis was performed by professional software named Similarity Evaluation System for Chromatographic Fingerprint of TCM, which was recommended by China Food and Drug Administration. The correlation coefficients of entire chromatographic profiles of the samples were calculated, and the similarities of different chromatograms are based on the cosine ratio between sample chromatogram and reference chromatogram. Hierarchical clustering analysis (HCA) was performed by SPSS 26.0 (SPSS, Chicago, IL, USA). Ward’s method, which is very efficient for the analysis of variance between clusters, was applied, and squared Euclidean distance was selected as measurement for the analysis. For further discrimination among samples, principal component analysis (PCA) pattern recognition method was employed using Soft Independent Modeling of Class Analogy-P 11.5 Demo.

## 4. Conclusions

In this study, a simple, accurate, and reliable UPLC-DAD-MS/MS method was developed and validated to evaluate the quality of PPP through fingerprint chromatogram combined with simultaneous determination of six types of components from sixteen analytes, namely flavonoids (11, 12, 13, and 16), monoterpenes and their glycosides (3, 4, 6, 14, and 15), tannins (7, 9, 10), phenolic acids (1, 2, and 5), and aromatic acids (8). Twelve samples were divided into two groups in HCA and PCA. In the quantitative determination, six types of components of the 12 varieties of PPP were successfully separated and determined. The content of flavonoids, tannins, and monoterpenes (≥19.34 mg/g) was the highest, accounting for more than 78.45% of the total compounds. The results demonstrated that the flavonoids, tannins, and monoterpenes were considered to be the key indexes in the classification and quality assessment of PPP. The UPLC fingerprint analysis is based on sixteen mainly bioactive constituents’ detection in different varieties of PPP samples combining SE, HCA, PCA, and quantification analysis can be effectively applied to evaluate the quality of the peony flower. The method proved to have good linearity, injection precision, repeatability, recovery, and sample stability. It is demonstrated that UPLC-DAD-MS/MS method coupled with multiple compounds determination and fingerprint analysis is a powerful, practical tool for comprehensive QC of peony flower. Thus, this study can be effectively used to evaluate the compounds in peony flower raw material for product quality assurance in the food, pharmaceutical, and cosmetic industries. Moreover, this study provides ideas for future research and the improvement of products by these industries. 

## Figures and Tables

**Figure 1 molecules-28-07741-f001:**
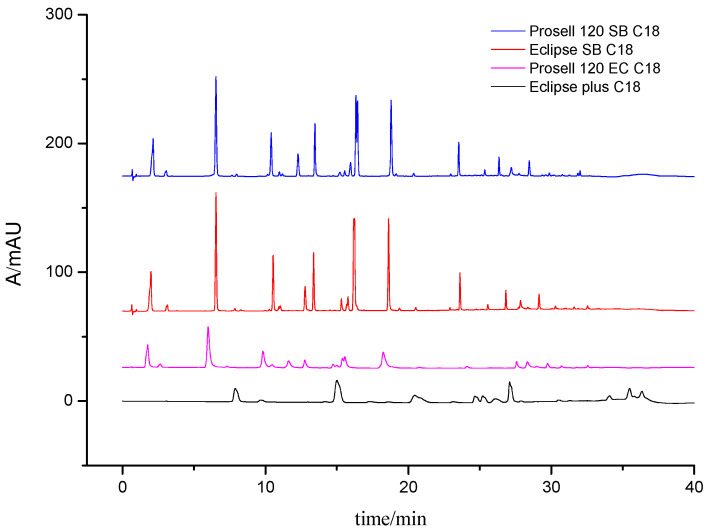
Comparison of different chromatographic columns.

**Figure 2 molecules-28-07741-f002:**
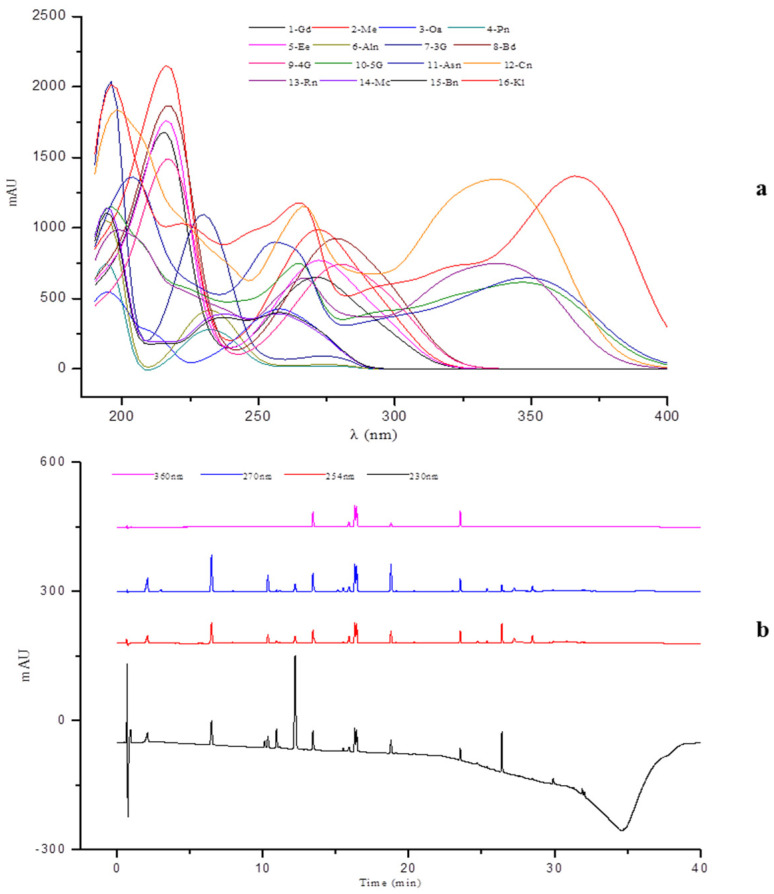
The UV spectra of mixed standards at different wavelengths. (**a**) the UV spectrogram of the sixteen components at a full wavelength scan (190–400 nm), (**b**) the DAD chromatograms of mixed standards at four different wavelengths).

**Figure 3 molecules-28-07741-f003:**
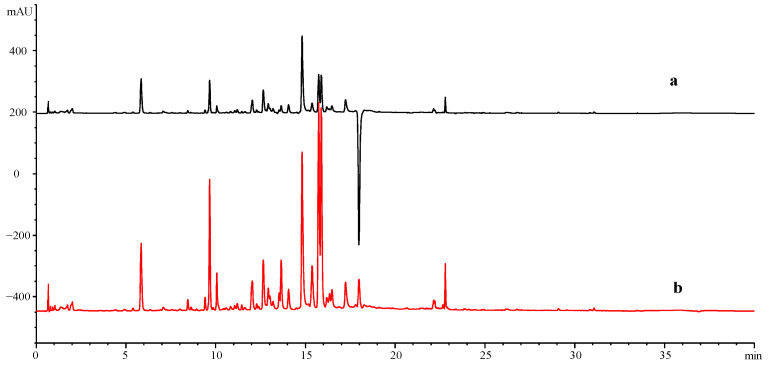
The DAD diagram of the sample (S8) at a wavelength of 270 nm. ((**a**): with reference, Ref = 360:100, (**b**): without reference, Ref = off).

**Figure 4 molecules-28-07741-f004:**
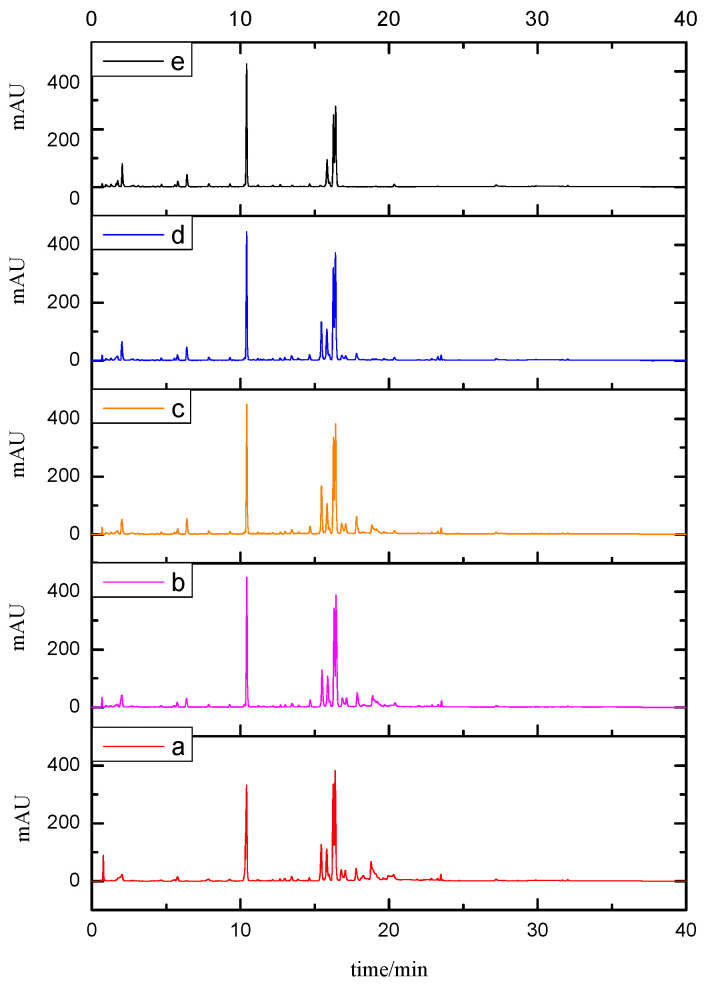
Comparison of different extraction solvents. ((**a**): 70% ethanol, (**b**): 30% methanol, (**c**): 50% methanol, (**d**): 70% methanol, (**e**): 90% methanol).

**Figure 5 molecules-28-07741-f005:**
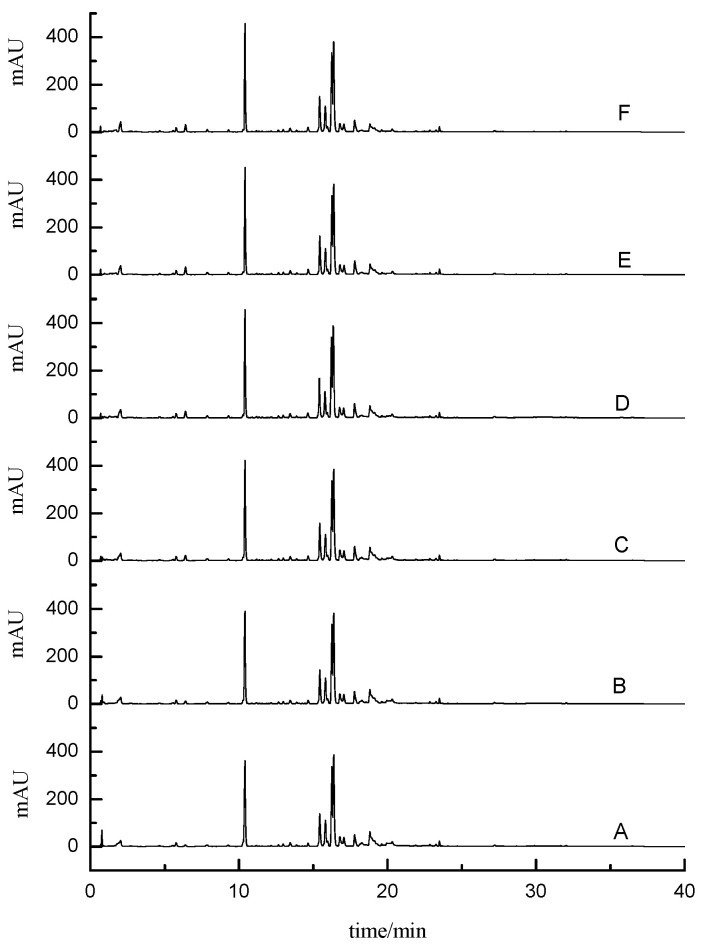
Comparison of the three mixed solutions of methanol/ethanol/water ((**A**)—1:6:3, (**B**)—2:5:3, (**C**)—3:4:3, (**D**)—4:3:3, (**E**)—5:2:3, (**F**)—6:1:3, *v*/*v*/*v*).

**Figure 6 molecules-28-07741-f006:**
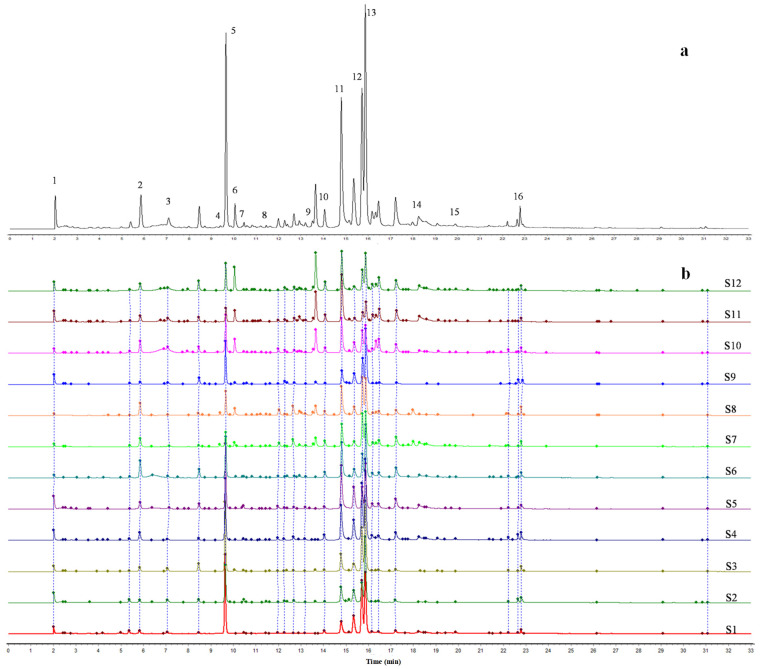
The UPLC-DAD chromatographic fingerprints of PPP extract. ((**a**): The reference characteristic spectrogram for chromatographic fingerprint, peak number: 1, Gd; 2, Me; 3, Oa; 4, Pn; 5, Ee; 6, Aln; 7, 3G; 8, Bd; 9, 4G; 10, 5G; 11, Asn; 12, Cn; 13, Rn; 14, Mc; 15, Bn; and 16, Kl; (**b**): The fingerprint chromatograms of 12 varieties of PPP samples which collected from Luoyang places of production is from chromatogram S1 to S12.)

**Figure 7 molecules-28-07741-f007:**
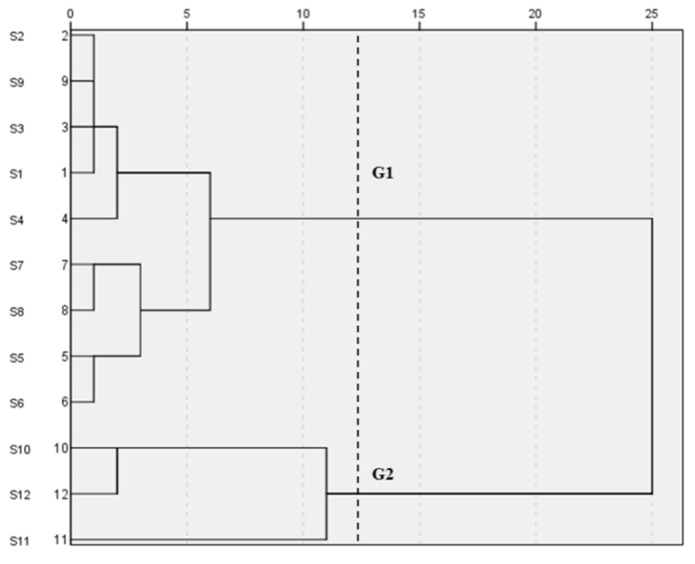
Dendrogram of HCA of 12 varieties PPP samples. “G” means group.

**Figure 8 molecules-28-07741-f008:**
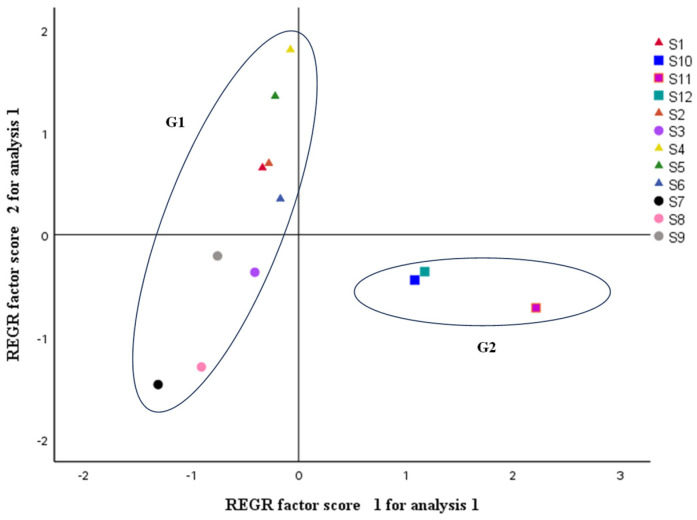
The scores plot obtained by PCA of the 12 varieties of PPP samples.

**Figure 9 molecules-28-07741-f009:**
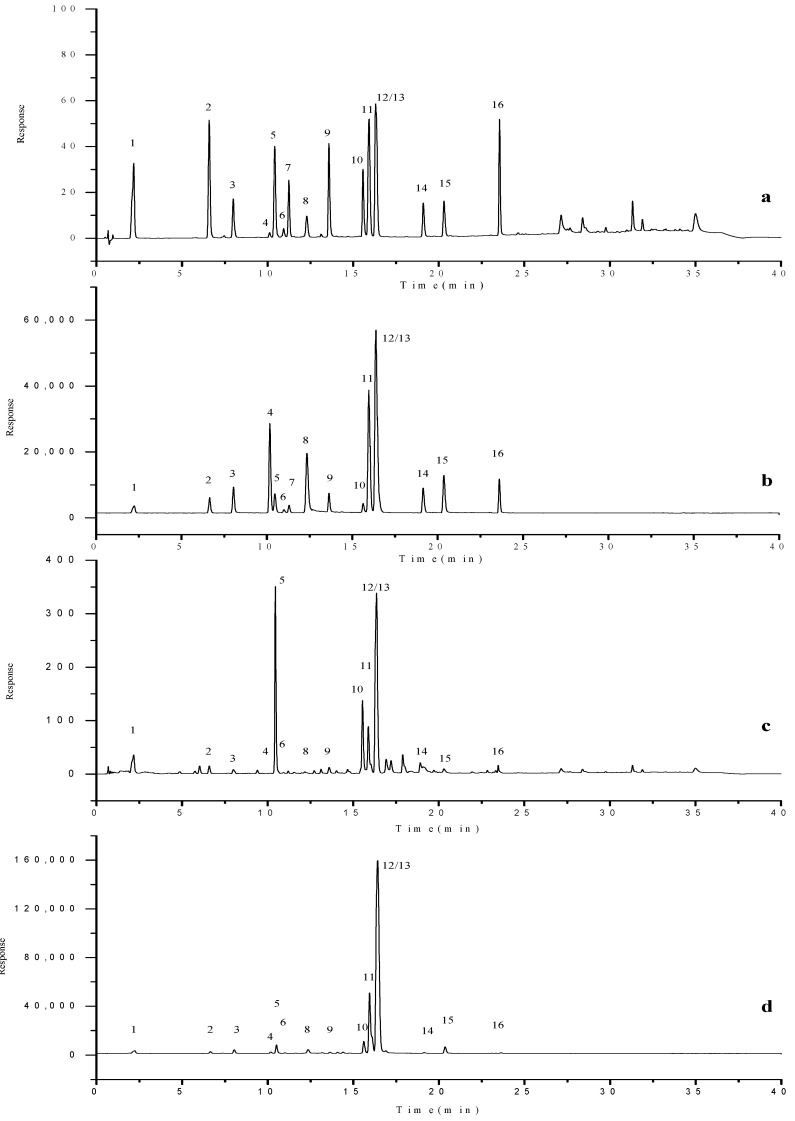
The DAD chromatogram (**a**,**c**) and total ion flow diagram (TIC, (**b**,**d**)) of mixed standard sample (**a**,**b**) and S1 sample (**c**,**d**) under multiple reaction monitoring mode (MRM). (1, Gd; 2, Me; 3, Oa; 4, Pn; 5, Ee; 6, Aln; 7, 3G; 8, Bd; 9, 4G; 10, 5G; 11, Asn; 12, Cn; 13, Rn; 14, Mc; 15, Bn; 16, Kl).

**Figure 10 molecules-28-07741-f010:**
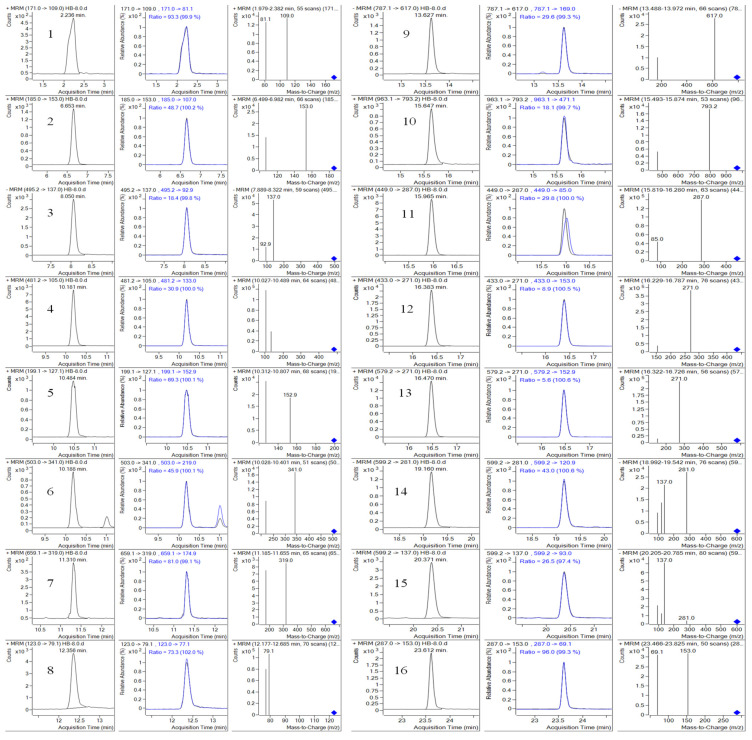
The MRM spectra of 16 target measured components. (1, Gd; 2, Me; 3, Oa; 4, Pn; 5, Ee; 6, Aln; 7, 3G; 8, Bd; 9, 4G; 10, 5G; 11, Asn; 12, Cn; 13, Rn; 14, Mc; 15, Bn; 16, Kl).

**Figure 11 molecules-28-07741-f011:**
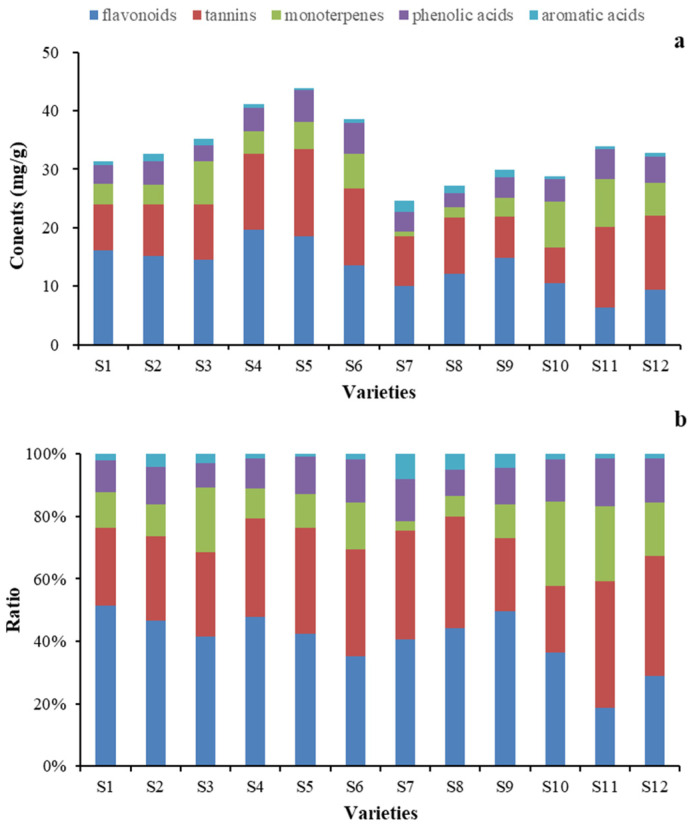
The bar graph of average contents of five types of active compound in PPP samples from 12 varieties. (**a**) the total contents of five types of active compound, (**b**) the ratio of every type of active compound in all five types of active compound.

**Figure 12 molecules-28-07741-f012:**
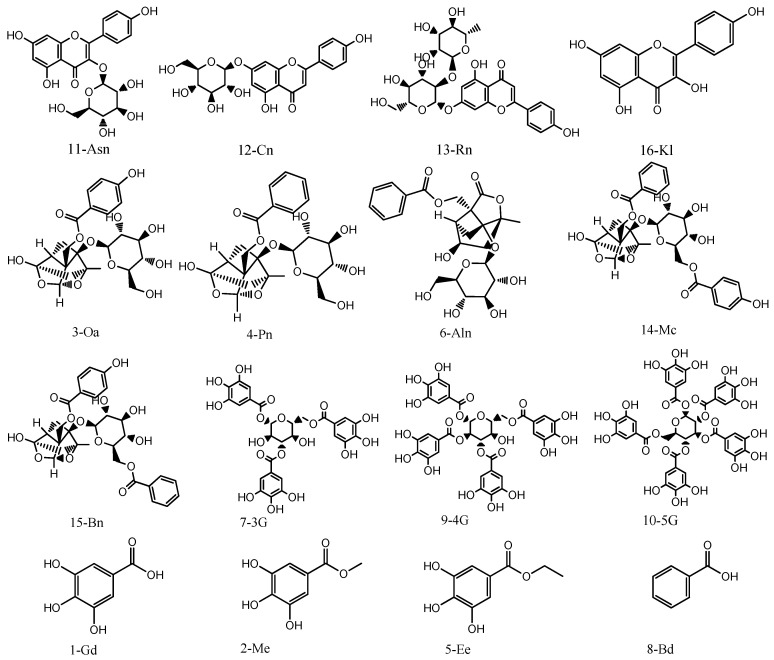
Chemical structures of the sixteen components including flavonoids (11, 12, 13, and 16), monoterpenes and their glycosides (3, 4, 6, 14, and 15), tannins (7, 9, and 10), phenolic acids (1, 2, and 5), and aromatic acids (8).

**Table 1 molecules-28-07741-t001:** Similarities from 12 varieties of PPP samples.

Number	Varieties												
	S1	S2	S3	S4	S5	S6	S7	S8	S9	S10	S11	S12	R
S1	1	0.905	0.978	0.953	0.857	0.846	0.827	0.849	0.946	0.744	0.459	0.653	0.923
S2	0.905	1	0.954	0.948	0.865	0.889	0.823	0.825	0.985	0.725	0.536	0.703	0.931
S3	0.978	0.954	1	0.967	0.861	0.889	0.848	0.868	0.979	0.754	0.5	0.697	0.946
S4	0.953	0.948	0.967	1	0.91	0.926	0.927	0.934	0.962	0.802	0.615	0.757	0.978
S5	0.857	0.865	0.861	0.91	1	0.937	0.826	0.853	0.869	0.86	0.741	0.793	0.944
S6	0.846	0.889	0.889	0.926	0.937	1	0.885	0.898	0.883	0.836	0.706	0.795	0.954
S7	0.827	0.823	0.848	0.927	0.826	0.885	1	0.974	0.837	0.833	0.74	0.84	0.936
S8	0.849	0.825	0.868	0.934	0.853	0.898	0.974	1	0.844	0.854	0.722	0.827	0.946
S9	0.946	0.985	0.979	0.962	0.869	0.883	0.837	0.844	1	0.743	0.524	0.709	0.943
S10	0.744	0.725	0.754	0.802	0.86	0.836	0.833	0.854	0.743	1	0.856	0.929	0.894
S11	0.459	0.536	0.5	0.615	0.741	0.706	0.74	0.722	0.524	0.856	1	0.935	0.737
S12	0.653	0.703	0.697	0.757	0.793	0.795	0.84	0.827	0.709	0.929	0.935	1	0.861
R	0.923	0.931	0.946	0.978	0.944	0.954	0.936	0.946	0.943	0.894	0.737	0.861	1

**Table 2 molecules-28-07741-t002:** The eigenvalues, percentage, and cumulative percentage of PCA.

Component	Extraction Sum of Squared Loadings		
	Total	Percent of Variance	Cumulative %
1	11.298	28.970	28.970
2	6.781	17.387	46.357
3	5.439	13.946	60.302
4	4.162	10.671	70.973
5	3.468	8.892	79.865
6	2.387	6.121	85.987
7	1.778	4.558	90.544
8	1.573	4.032	94.577

**Table 3 molecules-28-07741-t003:** Calibration curves and concentration ranges of sixteen analytes.

Analytes	Calibration Curves	Concentration Ranges (μg/mL)	*R*	LOD (ng/mL)	LOQ (ng/mL)
Gd	y = 0.5550x + 63.6139	0.21–21.19	0.9994	22.72	75.72
Me	y = 1.0436x + 311.6737	0.22–22.34	0.9993	3.02	10.08
Oa	y = 2.5998x + 1150.1327	0.19–19.19	0.9988	1.28	4.25
Pn	y = 3.2395x − 365.8647	0.41–41.08	0.9997	1.63	5.43
Ee	y = 1.1635x + 141.0442	0.21–21.03	0.9999	3.25	10.85
Aln	y = 0.0800x + 40.7541	0.39–39.44	0.9998	19.52	65.06
3G	y = 0.3953x − 49.1326	0.20–19.58	0.9999	13.52	45.05
Bd	y = 2.4119x + 1131.7842	0.41–40.96	0.9996	88.54	295.15
4G	y = 1.3849x − 345.0950	0.21–21.56	0.9996	6.07	20.24
5G	y = 0.8182x − 187.7223	0.20–20.06	0.9998	6.89	22.98
Asn	y = 6.7508x + 1519.9230	0.21–20.70	0.9994	0.75	2.50
Cn	y = 20.8332x + 1913.0233	0.20–19.99	0.9990	0.69	2.30
Rn	y = 12.0627x + 1565.9707	0.20–20.29	0.9995	0.64	2.13
Mc	y = 1.2153x + 3.6568	0.22–21.59	0.9991	3.98	13.28
Bn	y = 3.6202x + 808.4494	0.19–18.82	0.9996	9.14	30.46
Kl	y = 1.7154x + 254.3281	0.19–18.62	0.9998	6.00	20.01

**Table 4 molecules-28-07741-t004:** The precision test results (*n* = 6).

Analytes	Gd	Me	Oa	Pn	Ee	Aln	3G	Bd	4G	5G	Asn	Cn	Rn	Mc	Bn	Kl
Peak area (A)	1205	2687	6542	13,692	2887	390	748	11,287	2613	1463	15,629	45,031	27,350	2758	7958	3508
	1215	2689	6578	13,624	2878	384	753	11,197	2634	1468	15,546	45,145	27,486	2732	7966	3545
	1233	2644	6549	13,590	2870	387	762	11,302	2635	1452	15,456	45,127	27,374	2744	7994	3533
	1225	2656	6543	13,544	2853	393	755	11,100	2609	1445	15,357	45,084	27,326	2766	7943	3556
	1208	2632	6536	13,600	2892	388	750	11,456	2621	1454	15,389	45,021	27,273	2742	7937	3502
	1200	2668	6432	13,503	2832	381	746	11,285	2600	1466	15,423	45,043	27,160	2737	7920	3504
RSD (%)	1.04	0.86	0.77	0.48	0.79	1.10	0.76	1.05	0.53	0.62	0.66	0.12	0.40	0.47	0.32	0.66

**Table 5 molecules-28-07741-t005:** The repeatability test results (*n* = 6).

Analytes	Gd	Me	Oa	Pn	Ee	Aln	3G	Bd	4G	5G	Asn	Cn	Rn	Mc	Bn	Kl
Content (W/mg/kg)	2193	676	711	176	361	1649	15	619	361	7434	4131	4337	7588	162	860	77
	2187	666	708	172	359	1644	14	645	368	7647	4079	4458	7324	158	847	74
	2133	678	718	178	357	1650	14	632	359	7524	4210	4380	7434	160	832	72
	2146	680	720	182	349	1638	14	622	363	7467	4100	4312	7344	157	830	75
	2190	654	715	170	355	1649	15	625	356	7418	4156	4413	7548	162	856	77
	2230	663	702	174	351	1635	15	632	365	7470	4154	4386	7563	163	857	78
Mean ^a^ (W/mg/kg)	2180	669	713	175	355	1644	15	629	362	7494	4138	4381	7467	161	847	76
RSD (%)	1.62	1.51	0.95	2.29	1.36	0.39	2.72	1.47	1.21	1.12	1.12	1.19	1.55	1.42	1.55	3.14

Notes: ^a^
*n* = 6 mean of content.

**Table 6 molecules-28-07741-t006:** The recovery for the analysis of the sixteen bioactive compounds (*n* = 3).

Analytes	Background (μg)	Added (μg)	Measured (μg)	Mean ^a^ (%)	RSD ^b^ (%)
Gd	542.73	211.89	756.30	100.79	3.13
		423.78	970.11	100.85	2.42
		847.57	1400.23	101.17	2.75
Me	168.35	111.72	282.64	102.30	2.50
		223.44	393.11	100.59	3.10
		446.88	610.72	98.99	2.20
Oa	178.14	95.94	275.33	101.30	4.13
		191.88	371.12	100.57	3.63
		383.76	545.31	95.68	4.80
Pn	43.84	20.54	65.03	103.15	4.24
		41.08	88.10	107.73	4.41
		82.16	125.76	99.70	3.83
Ee	89.76	42.06	133.21	103.32	4.11
		84.11	175.06	101.41	5.21
		168.23	257.47	99.69	4.44
Aln	411.95	197.20	608.42	99.63	3.44
		394.40	810.11	100.95	4.12
		788.80	1197.16	99.54	2.16
3G	3.61	19.58	24.31	105.69	7.67
		39.16	46.16	108.64	3.99
		78.33	89.14	109.19	3.22
Bd	158.01	81.92	234.25	93.06	2.18
		163.84	312.46	94.27	3.07
		327.67	480.18	98.32	2.11
4G	90.69	43.12	132.33	96.56	3.30
		86.24	174.15	96.77	2.14
		172.48	257.01	96.43	4.25
5G	1883.81	802.59	2663.24	97.11	3.22
		1605.17	3456.12	97.95	1.80
		3210.35	5080.1	99.56	3.65
Asn	1035.02	413.90	1444.75	98.99	4.72
		827.81	1858.44	99.47	1.28
		1655.61	2690.17	99.97	2.70
Cn	1097.90	399.84	1480.13	95.60	0.89
		799.68	1891.28	99.21	2.45
		1599.36	2680.41	98.95	3.01
Rn	1862.14	811.44	2704.1	103.76	1.54
		1622.88	3500.78	100.97	1.38
		3245.76	5097.42	99.68	2.86
Mc	40.02	21.59	63.47	108.62	4.76
		43.17	88.04	111.22	5.03
		86.35	127.09	100.84	6.42
Bn	211.63	94.10	310.75	105.34	4.73
		188.20	400.49	100.35	5.33
		376.40	592.04	101.07	3.21
Kl	18.61	9.31	29.34	115.22	4.86
		18.62	38.45	106.53	3.72
		37.25	57.19	103.58	4.03

Notes: ^a^
*n* = 3 mean of recovery. ^b^ Relative standard deviations.

**Table 7 molecules-28-07741-t007:** The stability test results (*n* = 3).

Analytes	Time (h)	0	1	2	4	8	12	24
Gd	Content (mg/kg)	2170.91	2168.44	2167.98	2130.45	2146.27	2141.21	2127.23
	stability (%)	/	99.89	99.87	98.14	98.86	98.63	97.99
	RSD (%)	1.52	1.88	2.33	2.05	3.46	3.13	4.30
Me	Content (mg/kg)	673.38	671.76	674.62	680.12	677.43	665.43	663.4
	stability (%)	/	99.76	100.18	101.00	100.60	98.82	98.52
	RSD (%)	0.91	3.12	3.54	4.76	3.63	4.10	4.31
Oa	Content (mg/kg)	712.57	711.42	710.78	708.46	709.11	705.64	706.32
	stability (%)	/	99.84	99.75	99.42	99.51	99.03	99.12
	RSD (%)	0.72	1.49	2.45	2.26	3.42	3.67	3.16
Pn	Content (mg/kg)	175.37	175.42	174.44	173.49	174.01	172.64	170.46
	stability (%)	/	100.03	99.47	98.93	99.22	98.44	97.20
	RSD (%)	1.64	2.43	1.64	3.28	4.13	2.31	3.09
Ee	Content (mg/kg)	359.04	359.18	357.41	358.76	356.23	354.26	350.11
	stability (%)	/	100.04	99.50	100.38	99.29	99.44	98.81
	RSD (%)	0.60	1.57	1.38	2.46	2.43	3.47	4.16
Aln	Content (mg/kg)	1647.82	1644.33	1645.77	1646.21	1643.24	1641.19	1640.73
	stability (%)	/	99.79	99.88	99.90	99.72	99.60	99.57
	RSD (%)	0.19	0.88	1.46	3.25	2.15	3.44	3.62
3G	Content (mg/kg)	14.45	14.72	15.72	14.23	14.22	14.07	13.97
	stability (%)	/	101.87	108.79	98.48	98.41	97.37	96.68
	RSD (%)	2.25	3.48	4.16	3.75	4.06	5.55	6.57
Bd	Content (mg/kg)	632.06	633.24	630.72	631.40	630.12	628.45	624.64
	stability (%)	/	100.19	99.79	99.90	99.69	99.43	98.83
	RSD (%)	2.02	1.51	1.67	3.11	3.32	2.44	4.39
4G	Content (mg/kg)	362.78	365.24	364.48	362.13	363.46	360.11	358.83
	stability (%)	/	100.68	100.47	99.82	100.19	99.26	98.91
	RSD (%)	1.34	1.46	2.56	3.17	3.67	4.12	4.08
5G	Content (mg/kg)	7535.24	7548.89	7532.02	7530.12	7544.13	7520.13	7525.63
	stability (%)	/	100.18	99.96	99.93	100.12	99.80	99.87
	RSD (%)	1.42	2.23	2.78	1.26	1.75	2.13	1.08
Asn	Content (mg/kg)	4140.09	4152.12	4137.22	4133.12	4117.76	4110.21	4104.31
	stability (%)	/	100.29	99.93	99.83	99.46	99.28	99.14
	RSD (%)	1.60	0.64	1.06	2.07	2.22	1.02	3.21
Cn	Content (mg/kg)	4391.58	4394.25	4382.49	4380.31	4376.48	4372.47	4366.38
	stability (%)	/	100.06	99.79	99.74	99.66	99.56	99.43
	RSD (%)	1.39	0.94	1.23	1.54	2.27	3.25	3.60
Rn	Content (mg/kg)	7448.55	7433.87	7445.24	7423.18	7378.15	7345.68	7320.01
	stability (%)	/	99.80	99.96	99.66	99.05	98.62	98.27
	RSD (%)	1.78	0.13	0.47	1.34	1.06	1.98	2.34
Mc	Content (mg/kg)	160.09	158.11	159.42	157.46	158.23	156.20	155.78
	stability (%)	/	98.76	99.58	98.36	98.84	97.57	97.31
	RSD (%)	1.04	2.03	2.95	3.18	3.64	2.84	4.13
Bn	Content (mg/kg)	846.51	848.33	845.80	843.43	842.15	840.41	834.76
	stability (%)	/	100.22	99.92	99.64	99.48	99.28	98.61
	RSD (%)	1.62	2.43	2.31	3.20	1.24	1.46	2.01
Kl	Content (mg/kg)	74.45	74.77	74.32	73.47	73.23	72.49	72.01
	stability (%)	/	100.43	99.83	98.68	98.36	97.37	96.72
	RSD (%)	3.38	3.72	3.86	4.15	4.77	5.01	6.14

**Table 8 molecules-28-07741-t008:** Contents of sixteen analytes in PPP samples (*n* = 3).

Sample	Content (mg/g)															
	Gd	Me	Oa	Pn	Ee	Aln	3G	Bd	4G	5G	Asn	Cn	Rn	Mc	Bn	Kl
S1	2.19	0.68	0.71	0.18	0.36	1.65	TR	0.62	0.36	7.43	4.13	4.34	7.59	0.16	0.86	0.077
S2	3.16	0.30	1.43	0.26	0.45	1.25	0.024	1.33	0.81	8.02	2.94	2.16	9.52	0.032	0.40	0.56
S3	2.07	0.52	1.75	0.020	0.17	5.26	TR	1.06	0.45	9.00	1.46	3.93	9.06	0.059	0.19	0.11
S4	2.19	1.39	0.65	0.32	0.38	2.27	TR	0.63	1.22	11.65	4.99	4.77	9.48	0.088	0.62	0.43
S5	3.75	1.41	0.19	0.050	0.20	4.46	TR	0.32	0.70	14.12	7.69	2.52	7.85	0.0049	0.041	0.47
S6	1.37	3.34	0.83	0.085	0.62	4.60	TR	0.62	0.57	12.61	2.05	2.80	8.36	0.070	0.27	0.32
S7	1.33	1.55	0.24	0.016	0.42	0.42	TR	2.02	0.57	8.03	1.20	3.14	5.62	0.017	0.060	0.012
S8	0.98	1.10	0.31	0.035	0.24	1.41	0.033	1.36	0.63	9.00	1.93	3.84	6.25	0.0053	0.018	0.037
S9	2.96	0.33	0.90	0.012	0.19	2.17	0.028	1.34	0.58	6.43	2.64	2.80	8.65	0.035	0.10	0.74
S10	1.93	1.66	1.47	0.15	0.36	5.51	0.031	0.47	0.87	5.26	2.68	2.83	4.84	0.20	0.43	0.15
S11	3.80	1.19	1.68	0.13	0.23	5.61	0.044	0.49	1.23	12.50	0.72	1.32	4.31	0.059	0.65	0.035
S12	2.90	1.41	0.67	0.039	0.28	4.11	0.030	0.51	1.15	11.40	0.75	2.36	6.26	0.12	0.71	0.059

Notes: TR-Trace qualitative detection.

**Table 9 molecules-28-07741-t009:** Mass parameters used for the analytes’ detection.

Peak Numbers	Analytes	Precursor Ions (*m*/*z*)	Product Ions (*m*/*z*)	Residence Time (ms)	Fragmentation Voltage (V)	Collision Energy (eV)	Mode
1	Gd	171.0	109.0 *	7	88	14	ESI^+^
			81.1			22	
2	Me	185	107.0	7	83	22	ESI^+^
			153.0 *			14	
3	Oa	495.2	92.9	7	195	60	ESI^−^
			137.0 *			30	
4	Aln	481.2	105.0 *	7	90	28	ESI^+^
			133.0			32	
5	Ee	199.1	127.1 *	7	71	14	ESI^+^
			152.9			14	
6	Pn	503.0	219.0	7	110	34	ESI^+^
			341.0 *			30	
7	3G	659.1	174.9 *	7	180	30	ESI^+^
			319.0			22	
8	Bd	123.0	77.1	7	86	22	ESI^+^
			79.1 *			14	
9	4G	787.1	169.0	7	225	60	ESI^−^
			617.0 *			28	
10	5G	963.1	471.1	7	250	46	ESI^+^
			793.2 *			26	
11	Asn	449.0	85.0	7	109	30	ESI^+^
			287.0 *			14	
12	Cn	433.0	153.0	7	128	60	ESI^+^
			271.0 *			14	
13	Rn	579.2	271.0 *	7	125	18	ESI^+^
			152.9			70	
14	Mc	599.2	281.0 *	7	175	30	ESI^−^
			137.0			42	
15	Bn	599.2	137.0 *	7	200	34	ESI^−^
			93.0			66	
16	Kl	287	69.1 *	7	145	60	ESI^+^
			153.0			34	

Note: * quantizer ion.

## Data Availability

Data available on request due to restrictions. The data presented in this study are available on reasonable request from the corresponding author.

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
