# Peer review of "Quality Evaluation of Peony Petals Based on the Chromatographic Fingerprints and Simultaneous Determination of Sixteen Bioactive Constituents Using UPLC-DAD-MS/MS"

_molecules, 2023, doi:10.3390/molecules28237741_

Round 1

Reviewer 1 Report

Comments and Suggestions for Authors

Comments

The manuscript is a well-written report. The authors describe a methodology that stands as a base to evaluate different peony petals through the identification of their phenolic and terpenic compounds, establishing fingerprints of 12 different varieties of peonies. The objective is clear, and the results demonstrate that the UPLC-DAD-MS/MS method is stable, consistent, and accurate. The conclusions are supported by the reported results. In my opinion, this manuscript can be published in the journal Molecules. However, there are some recommendations:

1.      Flavonoids are phenolic compounds, so it is not necessary to separate them in the introduction.

2.      Line 118: the word “and” is repeated.

3.      The figures and table order are not correct. If the authors are going to leave Figure 1 and Table 1 at the end of the manuscript because they are part of the methodology, they have to name them with the number that corresponds.

Reviewer 2 Report

Comments and Suggestions for Authors

The manuscript entitled "Quality evaluation of Peony Petal based on the chromatographic fingerprints and simultaneous determination of sixteen bioactive constituents by UPLC-DAD-MS/MS" reports a study of the determination of secondary metabolites of peony flowers (Paeonia) using UPLC-DAD-MS/MS.  The chromatographic data was used as input to perform similarity evaluation, hierarchical cluster analysis, principal component analysis, and quantification analysis.  The authors identify flavonoids, monoterpenes, phenolic acids, and aromatic acids.  The structure of sixteen compounds was reported.

The reported results can be used in further studies. However, there are some points to clarify and increase the visibility.

1 - Abstract - line 24

Please, replace " (8)were" with "(8) were"

2 - Abstract - line 15

"determination of sixteen bioactive constituents"

Please, clarify

3 - Figure 1 is on page 22, please review the order of the figures.

4 - Figure 9. Please, perhaps the number of the samples could replace the symbols.

5 - Figure 9. Please improve the figure showing the G1 and G2 group in the plot, to relate the results of HCA.

6 - Please, the discussion of the sixteen compounds reported must be improved, mainly because the authors put in the abstract and title the word "bioactive".
